# EXTENDING FLEXIBILITY OF IMAGE CODING ENHANCEMENT FRAMEWORK FOR IoTs

## ABSTRACT

Neural image compression, necessary in various edge-device scenarios, suffers from its heavy encode-decode structures and inflexible compression level switch. The primary issue is that the computational and storage capabilities of edge devices are weaker than those of servers, preventing them from handling the same amount of computation and storage. One solution is to downsample images and reconstruct them on the receiver side; however, current methods uniformly downsample the image and limit flexibility in compression levels. We take a step to break up this paradigm by proposing a conditional uniform-based sampler that allows for flexible image size reduction and reconstruction. Building on this, we introduce a lightweight transformer-based reconstruction structure to further reduce the reconstruction load on the receiver side. Extensive evaluations conducted on a real-world testbed demonstrate multiple advantages of our system over existing compression techniques, especially in terms of adaptability to different compression levels, computational efficiency, and image reconstruction quality.

## 1 INTRODUCTION

The need for advanced lossy image compression is raised by the explosive development of edge devices equipped with high-resolution cameras, such as industrial-inspections (George et al., 2019), wildlife observation (wil, 2023), and autonomous driving (Ananthanarayanan et al., 2017). Neural-Network (NN) based compressor can satisfy this need, which outperform traditional image compression techniques like JPEG (Group, 1986) and BPG (Bellard, 2014). However, due to its heavy, symmetric encoding and decoding structure and inflexible compression rate adjustment, current NN-based methods have not yielded practical use on resource-constrained edge devices (Dasari et al., 2022).

Given the paucity of computational ability on edge devices in general (Fut, 2020; ope, 2018; aws, 2023; Li et al., 2023a;b), a huge gap would exist in the edge compression/decompress and transmission latency. As shown in Fig. 1a, encoding an image can take as long as 18 seconds on high-end devices like the NVIDIA Jetson TX2. Downsampling image size at the sender and restoring it on the receiver is one way to alleviate this problem (Yin et al., 2023; Cheng et al., 2024). However, these solutions usually employ super-resolution, which uniformly downsample and restore images to fixed sizes, lacking flexibility for dynamic and complex compression needs in real-world applications.

We take a fresh look at this problem and introduce Easz, a lightweight compression enhancement framework that operates efficiently at the edge-sender with near-zero computational demand, while also maintaining efficiency on the receiver. Easz is compatible with all existing compression algorithms. The intuition of Easz is an implicit assumption undermined in current solutions: the image need to be uniformly downsampled. Easz includes an erase-and-squeeze process, which relaxes this assumption by designing a conditional uniform-based sampler. This technique provides a more adaptable and fine-grained compression level but also loses the chance to employ efficient reconstruction through convolution or the fast Fourier transform techniques. We then propose a receiver-side lightweight transformer architecture for efficient, high-quality reconstruction of erased patches. This involves a two-stage image patchify process to limit the scope of attention correlation calculations and a four-layer transformer model for pixel-level local image reconstruction. As shown in Figure 1b, Easz surpasses both the NN-based compressor and the traditional compressor.

The key contributions of this paper are:

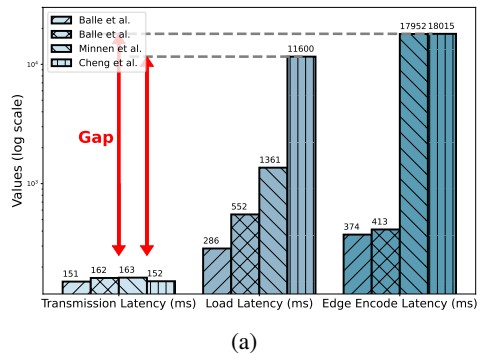 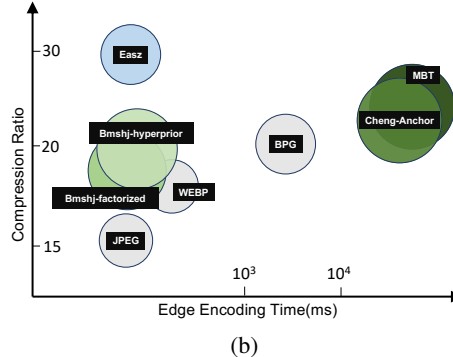

|      |      |
| :--: | :--: |
| (a)  | (b)  |

Figure 1: (a) NN-based compressors face challenges on edge devices like the Jetson TX2, where loading and encoding an image can take over 10 seconds compared to a transmission latency of about 0.1 seconds. (b) Easz is more efficient than other methods under the same image quality. Memory consumption is indicated by circle size; green circles represent GPU execution, while others indicate CPU execution.

- Generalized Erase-and-Squeeze Process: A new paradigm is introduced that offers more refined and flexible image reduction ratios;
- Receiver-Side Lightweight Transformer Architecture: A lightweight(8.7MB) transformer architecture is designed for efficient and high-quality reconstruction of erased patches;
- Compatibility with Existing Algorithms: Easz is compatible with all existing image compression algorithms and can also function independently;
- Enhanced Compression Flexibility and Efficiency: Easz offers significant compression flexibility and efficiency improvements. For the sender-side, Easz requires almost no additional computational cost with a controllable compression ratio, and on the reciever-side, Easz's reconstruction model is also lightweight, making it well-suited for real-world applications with varying and complex compression needs.

## 2 RELATED WORK

Learned-based image compression is experiencing significant growth, with advancements in end-to-end training, hyperprior structures, entropy models, and encoder-decoder improvements Notable developments include the introduction of auto-regressive components (Minnen et al., 2018), Gaussian Mixture Models for probability estimation (Cheng et al., 2020), and a general-purpose lossless compression paradigm using lightweight neural networks (Mao et al., 2022b;a; 2023). Attention mechanisms have been incorporated through Informer (Jun-Hyuk et al., 2022), while Transformers and Swin architectures are replacing traditional CNNs in encoding/decoding tasks (Yinhao Zhu and Cohen, 2022; He et al., 2021).

Despite progress, real-world applications still face challenges such as inflexibility in switching models and high latency at the edge. Deep-learning-based compression methods take about 1~20 second per image (512x768) on NVIDIA Jetson TX2, and many real-life endpoints are less potent than the TX2 (considering Raspberry Pi 4) but still need to compress images. A primary issue is that most NN-based image compressors require a model switch when changing compression levels. One approach involves downsampling images at the edge and using super-resolution techniques to reconstruct them on the server (Yin et al., 2023; Cheng et al., 2024). These methods reduce computational load at the edge, but applying super-resolution directly in this context results in an inflexible downsizing rate and can degrade reconstruction performance (Laroche et al., 2023; Jin et al., 2022).

## 3 SYSTEM DESIGN

The paper presents Easz, a novel edge-optimized image compression framework. It applies the erase-and-squeeze technique at the sender with a lightweight transformer-powered reconstruction

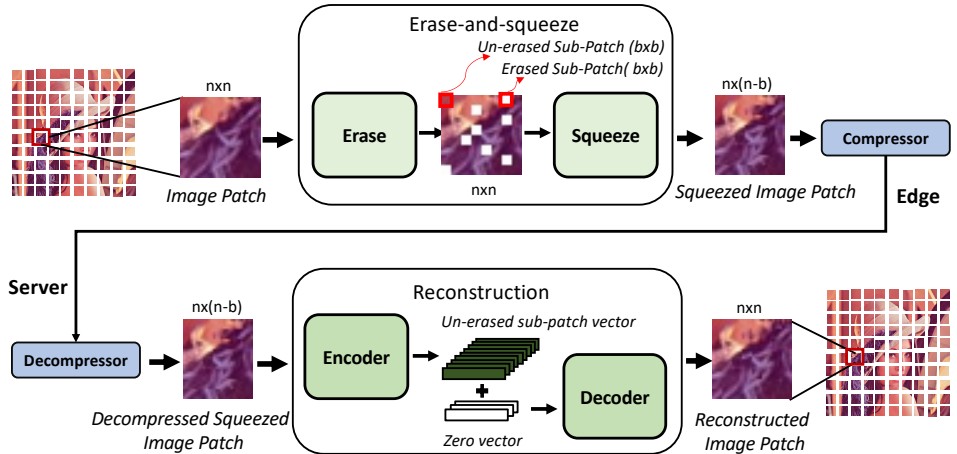

Figure 2: Easz system overview.

on the receiver side, outperforming conventional codecs like JPEG and other neural network-driven compressors. The default compressor used is JPEG due to its common use and prevalence. The whole framework is illustrated in Fig. 2. Next, we'll present our design step-by-step following the dataflow shown in Fig. 2. A detailed flexibility analysis is presented in §3.2.4.

## 3.1 IMPLICIT ASSUMPTION IN PREVIOUS METHODS

Previous image enhancement frameworks usually employ super-resolution as the downsample-reconstruction technique (Yin et al., 2023; Cheng et al., 2024). We point out that this introduces an implicit assumption and limits its flexibility.

The standard super-resolution (SR) model with multiple degradations typically posits that the low-resolution image is a degraded representation of a high-resolution image, characterized explicitly as a blurry, noisy, and sub-sampled version of the original.

$$y = (x \circledast k) \downarrow_s + \epsilon \quad \text{with} \quad \epsilon \sim \mathcal{N}(0, \sigma^2)$$

In this formulation, let $x$ denote the high-resolution image, $y$ represent its low-resolution counterpart, $k$ be the blur kernel, $\downarrow_s$ signify the subsampling operator with scale factor $s$, and $\epsilon$ denote the additive noise. *This model operates under the assumption that the blur kernel is uniform across the entire image, allowing for efficient computation of the low-resolution image through convolution or fast Fourier transform techniques*, as highlighted in recent studies (Laroche et al., 2023; Jin et al., 2022). However, when this model is directly implemented within an edge image-enhancement framework, the implicit assumption of uniformity introduces a constraint on the downsampling ratio, which in turn restricts the framework's flexibility. This limitation underscores the need for more adaptable techniques to accommodate varying degradation patterns, limiting the framework's flexibility.

Next, we will explain how to relax this assumption. The uniformly downsampled assumption is critical for applying efficient super-resolution-based reconstruction through convolution or fast Fourier transform techniques. By challenging this assumption, direct random pixel prediction becomes costly (see §3.3.2). We then propose a two-stage patchify process with a lightweight transformer to solve this problem.

## 3.2 ERASE-AND-SQUEEZE ALGORITHM

### 3.2.1 PROBLEM FORMULATION

Let $\mathbf{X} \in \mathbb{R}^{H \times W \times C}$ be a high-resolution image of an arbitrary size $H, W, C$. A sampler $G$ takes $\mathbf{X}$ as input and computes a downsampled image $\hat{\mathbf{X}} = G(\mathbf{X})$, where $\hat{\mathbf{X}} \in \mathbb{R}^{h \times w \times C}$.

Consider a coordinate system such that $\mathbf{X}[u, v]$ is the pixel value of $\mathbf{X}$ where $u, v \in [0, H - 1]$ and $[0, W - 1]$, respectively. $\hat{\mathbf{X}}[i, j]$ is the pixel value of $\hat{\mathbf{X}}$ at coordinates $(i, j)$ for $i \in \{1, 2, \ldots h\}, j \in \{1, 2, \ldots w\}$. Essentially, the sampler $G$ computes a mapping between $(i, j)$ and $(u, v)$. Practically, sampler $G$ contains two functions $\{g^0, g^1\}$ such that:

$$\hat{\mathbf{X}}[i, j] := \mathbf{X}\left[g^0(i, j), g^1(i, j)\right]$$

The uniform approach will have a sampler

$$G_u = \left\{g_u^0(i, j) = (i - 1)/(h - 1), g_u^1(i, j) = (j - 1)/(w - 1)\right\}.$$

### 3.2.2 CONDITIONAL UNIFORM-BASED SAMPLER

In this section, we aim to propose an effective sampler that challenges the implicit assumption discussed in Section 3.1. We treat each pixel as a sampling unit, though this can also be extended to patches (See §3.2.4). We initially introduce a Uniform-based sampler for row-based random sampling and subsequently impose constraints on it. Row-based sampling is employed to ensure that the sampled image can be reassembled into a rectangular format.

**Random Sampler Definition**. The random sampler $G_r$ computes a mapping between the coordinates $(i, j)$ of the downsampled image $\hat{\mathbf{X}}$ and the relative coordinates $(u, v)$ of the original image $\mathbf{X}$. In this sampler, each row $i$ of the image is processed sequentially, and within each row, the column coordinate $j$ is selected using a uniform random sampler. The sampler is defined by two functions, $g_r^1$, which governs the random column selection from $\mathbf{X}$, while $g_r^0(i)$ represents the current row:

$$\hat{\mathbf{X}}[i, j] = \mathbf{X}[g_r^0(i), g_r^1(i, j)]$$

where $g_r^0(i)$ is a deterministic function representing row selection, and $g_r^1(i, j)$ is a random mapping for the column coordinate within row $i$.

**Uniform Random Selection in Each Row**. To ensure uniform random sampling within each row of the original image $\mathbf{X}$, the row coordinate $g_r^0(i)$ is fixed as $i$, and the column selection function $g_r^1(i, j)$ samples a pixel uniformly from the width $W$ for each row $i$. The random sampler $G_r$ is described as follows:

$$G_r = \left\{g_r^0(i) = i, \quad g_r^1(i, j) = \text{Uniform}(0, W - 1)\right\}$$

Here, the row index $i$ remains fixed for each row, and the uniform random sampler $g_r^1(i, j)$ selects random column coordinates for each pixel $j$ in row $i$. This ensures uniform random selection across columns within each row while maintaining a structured row-based approach.

Applying this sampler directly results in poor compression ratios and reconstruction performance, as shown in Fig. 3(a). Quantitative analysis reveals that this issue stems from the adjacent sampled areas (see Fig. 3(b)). To address this problem, we introduce two constraints on the sampler.

**Constraints for Row-based Random Sampling.** When sampling from a matrix $X \in \mathbb{R}^{H \times W}$, where each row is sampled $T$ times, the new sample $x_{i, t+1}$ is subject to the following conditions:

1. Intra-row constraint (avoid proximity to previous samples in the same row):

$$g_r^1(i, t + 1) \sim \text{Uniform}(0, W - 1) \quad \text{subject to} \quad \left|g_r^1(i, t + 1) - g_r^1(i, t)\right| > \delta$$

Here, $\delta$ is a threshold distance that ensures the newly sampled column $g_r^1(i, t + 1)$ is sufficiently distant from the previously selected columns $\{g_r^1(i, 0), \ldots, g_r^1(i, t)\}$. This constraint guarantees a diverse selection of columns within each row, preventing the samples from clustering too closely together.

2. Inter-row constraint (minimize adjacency to prior samples from the preceding row):

$$g_r^1(i, t + 1) \sim \text{Uniform}(0, W - 1) \quad \text{subject to} \quad \left|g_r^1(i, t + 1) - g_r^1(i - 1, T)\right| > \Delta$$

Similarly, $\Delta$ represents a minimum separation between the newly sampled column in row $i$ and the previously selected columns $\{g_r^1(i - 1, 0), \ldots, g_r^1(i - 1, T)\}$ from row $i - 1$. This prevents adjacent

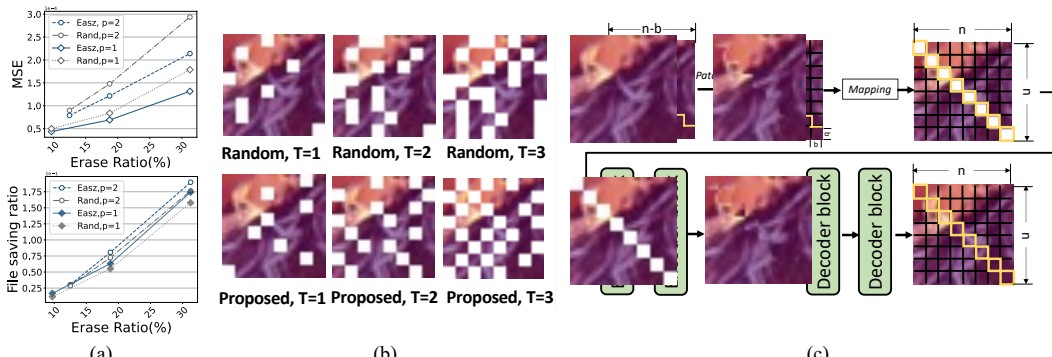

Figure 3: (a) The proposed method outperforms random masking in terms of JPEG impact and reconstruction, resulting in a higher file-saving ratio and lower MSE on Kodak dataset. The variable $p$ represents patch size. (b) Proposed erase methods compared with random erase methods. T indicates an erased item in each row. (c) Reconstruction process.

rows from sampling nearby columns, ensuring that the selection process avoids redundancy across rows.

Under these constraints, the row-based random sampler can be formalized as:

$$G_r = \left\{ g_r^0(i) = i, \quad g_r^1(i, t+1) = \begin{cases} \text{clip}\left(\text{Uniform}(0, W-1)\right) & \text{subject to} \\ |g_r^1(i, t+1) - g_r^1(i, t)| > \delta \\ |g_r^1(i, t+1) - g_r^1(i-1, T)| > \Delta \end{cases} \right\}$$

where the random column selection $g_r^1(i, t+1)$ is adjusted dynamically to satisfy both the intra-row and inter-row constraints. This ensures a well-distributed sampling process across the entire matrix, balancing randomness with structured diversity.

### 3.2.3 ERASE AND SQUEEZE

To handle the unsampled locations in the downsampled image $\hat{\mathbf{X}}$, we define a binary mask $\mathbf{M}$ of the same size as $\hat{\mathbf{X}}$ where each entry is set as follows:

$$\mathbf{M}[i, j] = 1 \text{ if } (i, j) \text{ is sampled, else } 0$$

The mask ratio, which determines the proportion of the image that is sampled, is controlled by the patch size $p$ and the sampled size $T$. Specifically, the choice of patch size influences the granularity of the sampling, while the sampled size $T$ dictates how many patches are included in the reconstruction for each row. By tuning these parameters, the framework can effectively balance the level of reconstruction difficulty and the computational load, enhancing model performance across various datasets. The mask will be sent with the compressed image for the receiver to decode.

The next step is to squeeze the non-zero (sampled) locations together to form a smaller image $\mathbf{X}_{\text{squeezed}} \in \mathbb{R}^{h' \times w' \times C}$, where $h' < h$ and $w' < w$ represent the dimensions of the squeezed image. This can be achieved by filtering out the zero entries from $\mathbf{X}$:

$$\mathbf{X}_{\text{squeezed}}[i', j'] = \mathbf{X}[i, j] \quad \text{for all } (i, j) \text{ where } \mathbf{M}[i, j] = 1$$

After erase-and-squeeze process, the squeezed image $\mathbf{X}_{\text{squeezed}}$ would be encoded using existing compressors like JPEG, BPG, etc to get a compressed form $\hat{\mathbf{X}}_{\text{squeezed}}$, as illustrated in Fig. 2.

### 3.2.4 FLEXIBILITY ANALYSIS

By relaxing the uniform sampling assumption outlined in §3.1, Easz can no longer utilize a super-resolution method for reconstruction, as the low-level continuous information becomes fragmented.

However, this shift creates new opportunities for flexibility. By controlling the sample size $T$, Easz can provide a more adaptable and fine-grained compression level compared to directly applying super-resolution techniques.

**Erase level.** Given an image with dimensions $H \times W \times C$, the overall compression ratio can be understood as comprising two components: 1) The ratio achieved through image size reduction. 2) The ratio obtained from the subsequent compression algorithm. The image size reduction is primarily controlled by the sampling size $T$ applied to each row. Thus, the reduction ratios can be expressed as $\frac{1}{W}, \frac{2}{W}, \frac{3}{W}, \ldots$. It's important to note that this proposed method can be similarly transposed to other axes, such as the columns.

So far, our discussion has focused on pixel-level sampling. However, the sample-erase-squeeze unit can be extended to operate on *patches*. By adopting this extension, we introduce another parameter that influences the reduction ratio: the patch size $p$. Consequently, experiments conducted with varying patch sizes will be detailed in §4.4. Under this patch-level sampling framework, the reduction ratios would be expressed as $\frac{p^2}{W}, \frac{2p^2}{W}, \frac{3p^2}{W}, \ldots$. In contrast to traditional super-resolution methods, which typically offer a single reduction pattern for a model, our sampler provides significant flexibility, enabling adaptation to various real-world applications.

**Model switching and mask transferring.** Switching models and transferring masks might be burdensome. As would introduced in §3.3.2, we present a lightweight transformer-based process capable of performing direct pixel-level reconstruction. This process is designed to handle any erase ratio since it is trained under this setting. Consequently, there is no need to prepare a separate model for each erase ratio or to switch models during compression ratio adjustments.

Another consideration is mask transferring. In our design, the mask is applied to small sub-patches (discussed in §3.3.2) created through a proposed two-stage image patching process. For instance, if the sub-patch size is $32 \times 32$, then the corresponding mask size would be $128$ bytes. This same mask is used for all sub-patches. Thus, the transmission of this size is not a concern.

As mentioned in the beginning, by utilizing the proposed erase-and-squeeze technique, the blur-kernel-based super-resolution method is no longer suitable for reconstruction. We then introduce a lightweight transformer architecture to directly conduct pixel prediction to address this issue.

## 3.3 RECONSTRUCTION

Given the squeezed compressed image $\hat{\mathbf{X}}_{\text{squeezed}}$, we introduced a Masked-Image-Modeling process to perform the pixel-level reconstruction on the receiver side. The reconstruction of the squeezed image back to the original image is approached through a framework reminiscent of masked image modeling (MIM). Specifically, the input is $\hat{\mathbf{X}}_{\text{squeezed}}$. $\mathbf{Y} = \mathcal{T}(\hat{\mathbf{X}}_{\text{squeezed}})$. The autoencoder model $\mathcal{G}(\cdot)$ takes the corrupted images $\hat{\mathbf{X}}_{\text{squeezed}}$ as input and aims to generate a prediction $\hat{\mathbf{Y}}$, optimizing the model by minimizing the difference between the prediction and the target:

$$\hat{\mathbf{Y}} = \mathcal{G}(\hat{\mathbf{X}}_{\text{squeezed}}), \quad \mathcal{L} = \mathcal{D}(\mathbf{X}, \hat{\mathbf{Y}})$$

This paper focuses on parametric target generation strategies utilizing a transformer structure for image reconstruction tasks. A significant challenge arises from directly predicting pixel values using transformers due to the quadratic complexity of attention mechanisms.

### 3.3.1 COMPLEXITY ANALYSIS.

Predicting pixel values for each element in the image, particularly in high-resolution images, proves to be computationally expensive. Let the transformer model's attention mechanism operate with complexity $\mathcal{O}(n^2 \cdot d)$, where $n$ is the number of tokens (patches or pixels) and $d$ is the dimensionality of the token embeddings. This results in costly computation when $n$ is large, especially for pixel-level operations.

For example, a 256x256 grayscale image would require $4,294,967,296 \times d_{model}$ calculations if each pixel were treated as a token. By employing the two-stage patchify process (with $n = 32$ and $b = 1$), this number is reduced by 256 times to $16,777,216 \times d_{model}$ calculations. Detailed analysis

would be in Appendix B. The reduction in complexity comes from performing attention operations within each patch rather than across the whole image.

### 3.3.2 Two-Stage Patchifying and Lightweight Restoration.

Motivated by this observation, we adopt a *two-stage patchifying process* followed by a lightweight transformer to improve computational efficiency.

1. First Stage: Initial Image Patchifying: Given the image $\mathbf{X} \in \mathbb{R}^{h \times w \times C}$, we divide the image into non-overlapping patches of size $p_h \times p_w$. For each patch $P_{m,n}$, the extraction follows:

$$P_{m,n} = \mathbf{X}[m \cdot p_h : (m+1) \cdot p_h - 1, n \cdot p_w : (n+1) \cdot p_w - 1, :]$$

where $m = 0, \ldots, \left\lfloor \frac{h}{p_h} \right\rfloor - 1$ and $n = 0, \ldots, \left\lfloor \frac{w}{p_w} \right\rfloor - 1$.

2. Second Stage: Subdividing into Sub-Patches: Each patch $P_{m,n}$ is further divided into sub-patches of size $S_h \times S_w$ to facilitate fine-grained prediction:

$$P_{m,n,k,l} = P_{m,n}[k \cdot S_h : (k+1) \cdot S_h - 1, l \cdot S_w : (l+1) \cdot S_w - 1, :]$$

where $k = 0, \ldots, \left\lfloor \frac{p_h}{S_h} \right\rfloor - 1$ and $l = 0, \ldots, \left\lfloor \frac{p_w}{S_w} \right\rfloor - 1$.

3. Lightweight Transformer for Sub-Patch Restoration: A lightweight transformer model, denoted as $\mathcal{T}_{\text{light}}(\cdot)$, with complexity $\mathcal{O}(m^2 \cdot d)$ (where $m$ is the number of sub-patches), is employed for restoring the missing pixel values within each sub-patch. Fig. 3(c) illustrates the architecture of our efficient transformer-based network on the server side. The encoder and decoder are composed of two transformer blocks, each containing three layernorms, one attention layer, and one feedforward layer. The model size is significantly reduced to $8.4$MB. Given the downsampled input sub-patch $P_{m,n,k,l}$, the predicted sub-patch $\hat{P}_{m,n,k,l}$ is generated as:

$$\hat{P}_{m,n,k,l} = \mathcal{T}_{\text{light}}(P_{m,n,k,l})$$

Through this reconstruction process, the original content of the image is effectively restored from the masked and squeezed representation. We adopt LPIPS (Zhang et al., 2018), a well-known perceptual loss, along with L1 loss as training loss as $\mathcal{L}$. Note that the encoder and decoder can work with a variety of input erase ratios, which is controlled by $p$ and $T$ (See §3.2.4), and hence, we do not need to train a separate model for each erase ratio. Sub-patches can execute in parallel both for encoder and decoder due to the nature of transformer block (Vaswani et al., 2017).

## 4 Experiments

### 4.1 Experimental Setting

**Training setting.** The experiments consist of two phases: offline pretraining and online testing. In the pretraining phase, a specific loss function is used with these hyperparameters: learning rate of 2.8e-4, erase ratio of 0.25, batch size of 4096, and weight decay of 0.05. Randomly generated erase masks are applied for model robustness during this stage. A consistent mask is utilized for online testing on both edge and server sides.

**Hardware platforms.** Our framework is implemented with $\sim$1000 lines of Python. We use an NVIDIA Jetson TX2 as the edge device and a desktop with Intel i7-9700K CPU and RTX 2080Ti GPU as the server, which are connected to a Wi-Fi router and communicate via TCP.

**Datasets.** During the offline pretraining phase, the CIFAR-10 (Alex and Geoffrey, 2009) dataset is employed to pre-train the model, enabling it to acquire generative capabilities. In the testing phase, two common image compression datasets, Kodak (Company, 1993) and CLIC (Workshop and on Learned Image Compression, 2022), are utilized to assess the generative performance of the proposed method.

**Metrics.** Since removing content would negatively impact reference-based metrics such as PSNR and SSIM (a trend also observed in other downsampled-and-super-resolution methods), we employ non-reference perceptual metrics for comparison with different compression techniques: Brisque (Mittal

et al., 2012), Pi (Blau et al., 2018), and Tres (Golestaneh et al., 2022). To benchmark against other super-resolution approaches, we include PSNR and SSIM to demonstrate the superiority of our method. Furthermore, we have conducted image classification experiments on the reconstructed images to showcase the proposed compression method's robustness in handling image analytic tasks. Compression performance is evaluated by bits per pixel (BPP).

**Baselines.** We use four compression methods as baselines to demonstrate the effectiveness of the proposed method: JPEG, BPG, MBT(Minnen et al., 2018), and Cheng-Anchor. Among these, MBT and Cheng-Anchor are two neural-network-based compression methods.

## 4.2 LATENCY ANALYSIS AND RESOURCE CONSUMPTION

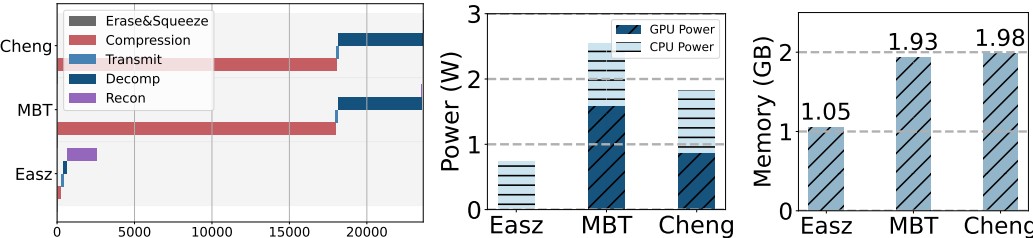

(a) End-to-End Latency Breakdown(ms).  (b) Encode Power consumption.  (c) Encode Memory footprint.

Figure 4: Efficiency Evaluation on NVIDIA Jetson TX2.

We first report the latency breakdown of Easz with other neural network-based compression methods, using a Jetson TX2 for compression and a server for decompression. We repeat the runs 24 times and report the average on Fig. 4a. We observe that Erase-and-Squeeze only takes up 0.7% of the end-to-end latency, which induces minimal overhead on the edge device and proves the efficiency of Easz' design. While both MBT and Cheng-Anchor are too compute-intensive to run the compression on the edge side. As expected, the reconstruction in Easz takes the longest time, accounting for 74% of the latency. We argue that it can be significantly improved by upgrading to a datacenter-class GPU, such as the A100, instead of the RTX 2080Ti.

Resource consumption is another critical consideration when it comes to resource-constrained edge devices. To assess this, we measure three key metrics – CPU power, GPU power, and memory footprint – using the Tegrastats Utility (teg, 2023) on the Jetson TX2. As illustrated in Fig. 4, our findings reveal that Easz excels in all metrics compared to other NN-based compression methods. Specifically, in contrast to MBT and Cheng-Anchor, Easz achieves a remarkable 71.3% and 59.9% reduction in total power consumption. It's noteworthy that Easz does not utilize any GPU power on the edge device, attributed to its lightweight yet effective erase-and-squeeze design. Furthermore, Easz reduces memory footprint by 45.8% and 47.1%, respectively. These results underscore the advantage of deploying Easz on wimpy edge devices.

## 4.3 COMPARISON WITH SUPER-RESOLUTION METHODS

We compare the reconstruction effect of Easz with state-of-the-art super-resolution methods to demonstrate Easz's effectiveness. As shown in Tab. 1, Easz outperforms super-resolution in pixel-level reconstruction metrics while having a much more flexible reduction ability. Note that Easz uses a model of only 8.7MB, while other models are 67MB.

Table 1: Comparison with Super-Resolution on Kodak Dataset.

| Metrics | Easz | SwinIR | realESRGAN | BSRGAN |
|---|---|---|---|---|
| PSNR | 28.96 | 24.86 | 24.85 | 25.35 |
| MS_SSIM | 0.96 | 0.94 | 0.93 | 0.94 |
| Recon Model Size | 8.7MB | 67MB | 67MB | 67MB |

Experimental results demonstrate that Easz surpasses traditional super-resolution in terms of PSNR and SSIM metrics when reducing pixel count equivalently. Figure 5 illustrates a comparison of image detail reconstruction, where Easz and other super-resolution all perform 2x reconstruction. It is evident that Easz better preserves image details; the children's faces are clearer, and the characters

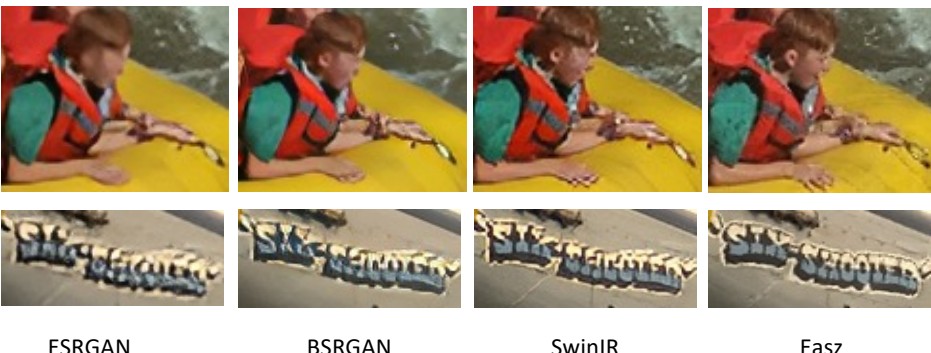

ESRGAN        BSRGAN        SwinIR        Easz

Figure 5: Reconstruction comparison of Easz with super-resolution methods. Easz better preserves image details due to its direct pixel value prediction, resulting in improved PSNR and SSIM.

are more recognizable. In contrast, the super-resolution reconstructed image is unsatisfactory. More quantitative results are shown in Appendix E

## 4.4 ABLATION STUDY

**Effectiveness of proposed sampler.** Fig. 6a and Fig. 6b compares the proposed erase mask generation method, the random erase mask method, and the baseline(JPEG and BPG) throughout the entire pipeline. It can be observed that the proposed erase mask generation method achieves better BPP at the same quality level on both JPEG and BPG, further substantiating the effectiveness of the proposed method.

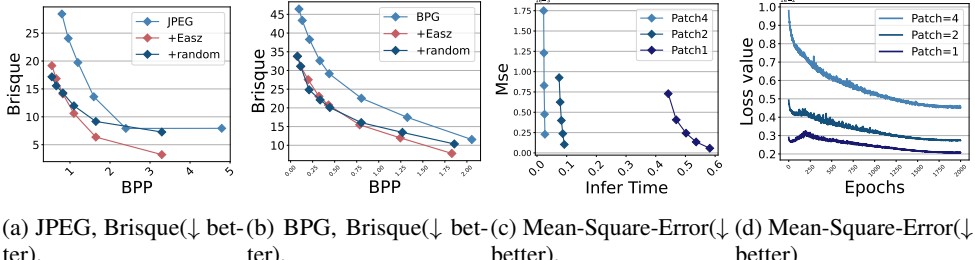

(a) JPEG, Brisque(↓ bet- (b) BPG, Brisque(↓ bet- (c) Mean-Square-Error(↓ (d) Mean-Square-Error(↓
ter).                  ter).                 better).          better)

Figure 6: (a)(b): Comparison between Easz with proposed mask strategy, Easz with random mask strategy, and conventional compression baselines (JPEG and BPG). (c) Patch size and erase ratio's impact on MSE(↓ better). (d) MSE(↓ better) during Easz fine-tuning process with patch size=1,2,4 on Kodak dataset.

**Patch size selection.** Fig. 6c examines the effects of two hyperparameters, erase block size (1, 2, and 4) and erase ratio (10% to 50%), on compression rate and quality. As the erase ratio increases, MSE rises, indicating lower reconstruction quality. Smaller patch sizes yield better reconstruction due to higher local correlations. Patch size=2 offers a balance between speed—being six times faster than size=1—and quality—with only a slight difference in MSE. Doubling the patch size from 2 to 4 also doubles both speed and MSE. The recommendation is to use smaller patch sizes for practical applications but consider size=2 for additional speed needs.

**Effectiveness of fine-tuning.** Our model, after pretraining on the CIFAR dataset for 5000 epochs, can be applied to various image compression tasks due to its ability to recognize similarities in local image features. Typically, models are first pre-trained on a large dataset and then fine-tuned for specific tasks. We tested if fine-tuning our pretrained model with the Kodak dataset would be beneficial and found that it indeed improves performance by reducing losses across different patch sizes (1x1, 2x2, and 4x4), as shown in Fig. 6d. This suggests that online fine-tuning of pre-trained models could further enhance compression effectiveness in real-world applications.

## 4.5 Improvement on Existing Compressors

To evaluate how well Easz works with leading compressors, we incorporated it into four established methods: JPEG and BPG (traditional compressors), as well as MBT and Cheng-anchor (neural network-based compressors). We used two datasets, Kodak and CLIC, to test the resilience of Easz across different types of image data. For the Kodak dataset, we aimed for a bit-per-pixel (BPP) rate of approximately 0.4; for the CLIC dataset, we targeted a BPP of around 0.3 to gauge Easz's efficacy at varying levels of compression. The results showing how each baseline method performs on its own and when combined with Easz —are detailed in Tables 2.

Table 2: Compression Performance Enhancement on Kodak Dataset and Clic Dataset.

| Metrics | | JPEG | | BPG | | MBT | | Cheng-anchor | |
|---|---|---|---|---|---|---|---|---|---|
| | | *Org* | *+Proposed* | *Org* | *+Proposed* | *Org* | *+Proposed* | *Org* | *+Proposed* |
| Kodak | BPP | 0.412 | 0.411 | 0.382 | 0.410 | 0.433 | 0.389 | 0.418 | 0.402 |
| | Brisque | 43.06 | 22.34 | 30.675 | 23.27 | 28.13 | 18.63 | 29.16 | 20.51 |
| | Pi | 4.84 | 3.33 | 3.07 | 3.04 | 3.01 | 3.00 | 3.11 | 3.05 |
| | Tres | 77.62 | 86.26 | 83.55 | 85.88 | 84.14 | 88.03 | 88.53 | 89.80 |
| Clic | BPP | 0.306 | 0.307 | 0.308 | 0.293 | 0.308 | 0.292 | 0.287 | 0.267 |
| | Brisque | 60.51 | 23.63 | 39.95 | 25.27 | 32.20 | 18.37 | 35.42 | 21.55 |
| | Pi | 8.51 | 5.02 | 4.85 | 4.66 | 4.33 | 4.35 | 4.58 | 4.50 |
| | Tres | 50.65 | 63.69 | 65.14 | 67.08 | 73.54 | 78.30 | 82.91 | 83.95 |

## 4.6 End-to-End Compression Performance

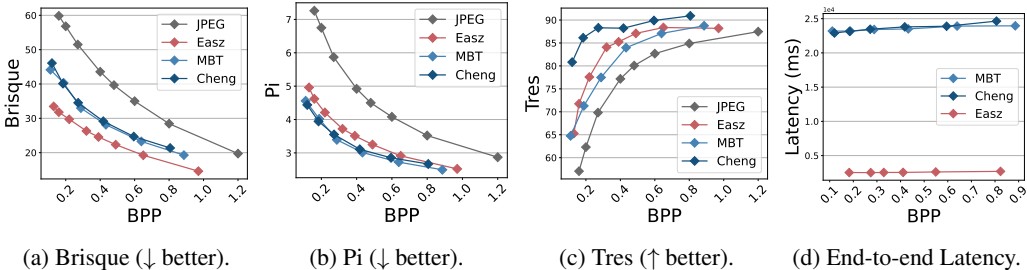

(a) Brisque (↓ better).  (b) Pi (↓ better).  (c) Tres (↑ better).  (d) End-to-end Latency.

Figure 7: Compression performance of Easz, JPEG, MBT and Cheng on three perceptual metrics (a-c). Fig. 7d evaluates the end-to-end latency on our testbed.

In this experiment, we use JPEG+Easz as the baseline and observe changes in three perceptual metrics at different bitrates (BPP). Notably, JPEG alone underperforms compared to two deep-learning compression methods at all compression levels. However, with Easz enhancement, JPEG shows a marked improvement. For the BRISQUE metric specifically, JPEG+Easz exceeds both deep-learning methods. Regarding the Pi metric, JPEG+Easz matches the performance of these methods. With the Tres metric, while JPEG+Easz outdoes MBT, it falls short of Cheng-anchor's results. Overall, Easz boosts JPEG to compete effectively with other state-of-the-art deep-learning compression techniques in each perceptual measure. Easz also outperforms two neural network-based methods in latency, with an average end-to-end latency of 2568ms across different bitrates per pixel, marking an 89% reduction compared to MBT and Cheng's methods.

## 5 Conclusion

This paper proposes Easz, which addresses the challenges of edge image compression and transmission latency by introducing a novel erase-and-squeeze technique that enhances flexibility and efficiency. By relaxing the conventional requirement for uniform downsampling, Easz allows for adaptable compression levels tailored to dynamic real-world applications. The implementation of a lightweight transformer architecture on the receiver side ensures high-quality reconstruction of erased image patches without imposing significant computational demands. Moreover, Easz's compatibility with existing compression algorithms makes it a versatile solution for modern edge devices. Our real-world evaluation in an edge-server testbed demonstrates Easz's improvement in compression performance and efficiency, emphasizing its potential for real-world applications.

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

## A  Detailed Efficiency Report for NN-based Compressors

To further illustrate the challenges faced when deploying current neural network-based compressors on edge devices, this section provides a detailed report on the performance of the four compression methods: b-facDailan et al. (2017), b-hyperBallé et al. (2018), MBTMinnen et al. (2018), Cheng-AnchorCheng et al. (2020) on actual edge devices. This area has been seldom explored in previous research. Table 3 shows the Edge FLOPs per image (512x768), model size for a single compression level, and loading time for five deep learning compression methods on NVIDIA Jetson TX2. It is important to note that the model size is only for one compression level; therefore, actual storage requirements are calculated by multiplying the given number by the number of compression levels. For example, Cheng-Anchor has six levels of compression, thus requiring a total storage space of 110MB*6=660MB. It's worth mentioning that although quantization can improve both space occupancy and operational efficiency of models, this approach involves performance, and it's well-known that quantization presents issues: its deployment on edge devices is complex and difficult to standardize. In contrast, Easz offers an easily deployable framework.

Table 3: Details of representative NN-based compression methods. Note that the loading and compression times are measured on an NVIDIA Jetson TX2, and FLOPs are evaluated with (512×768) image size.

| NN-based methods | b-fac | b-hyper | MBT | Cheng-Anchor |
|---|---|---|---|---|
| Edge-FLOPs | 36G | 36G | 36G | 145G |
| Model size (MB) | 28 | 46 | 118 | 110 |
| Loading time (ms) | 286 | 552 | 1361 | 1600 |
| Compression time (ms) | 374 | 413 | 17952 | 18015 |

## B  Two-Stage Image Patchifying Analysis

Consider a grayscale image of size $256 \times 256$ with a patch size of $1 \times 1$. This results in $256 \times 256 = 65,536$ pixels. In an attention-based model, treating each pixel as a token, the computational complexity for self-attention is given by:

$$O((h \times w)^2 \times d_{\text{model}})$$

where $h$ and $w$ are the height and width of the image, and $d_{\text{model}}$ is the model dimension. For a $256 \times 256$ image, the calculation becomes:

$$O(65536^2 \times d_{\text{model}}) = O(4,294,967,296 \times d_{\text{model}})$$

This level of computation is prohibitively expensive on modern hardware, and the complexity increases rapidly for higher-resolution images.

To reduce the computational load, we employ a two-stage patchify process. The first stage divides the image into non-overlapping patches of size $n \times n$. The number of patches is:

$$\frac{h \times w}{n^2}$$

Each patch is treated as a token, and the complexity of attention at this stage is reduced to:

$$O\left(\left(\frac{h \times w}{n^2}\right)^2\right) = O\left(\frac{(h \times w)^2}{n^4}\right)$$

This reduces the number of tokens, thus lowering the complexity. However, further refinement can be achieved with a second stage of patchification.

Each $n \times n$ patch is subdivided into $b \times b$ sub-patches in the second stage. The number of sub-patches is:

$$\frac{n^2}{b^2}$$

This results in a total of:

$$\frac{h \times w}{b^2}$$

sub-patches, and the attention complexity at this stage becomes:

$$O\left(\frac{h \times w}{n^2} \times \frac{n^4}{b^4}\right) = O\left(\frac{(h \times w) \times n^2}{b^4}\right)$$

Thus, the complexity is much lower than the original full-image attention, even for very small sub-patches (e.g., when $b = 1$). The final complexity is:

$$O(h \times w \times n^2)$$

For the example of a $256 \times 256$ image, where $n = 32$ and $b = 1$, the complexity reduces from:

$$O(65536^2 \times d_{\text{model}}) = O(4,294,967,296 \times d_{\text{model}})$$

to:

$$O(16,777,216 \times d_{\text{model}})$$

This represents a 256-fold reduction in computational complexity compared to the original.

## C    INFERENCE FOR EASZ

During the inference stage, the procedure starts from receiving a set of un-erased sub-patches $U = \{u_1, u_2, \ldots, u_m\}$, and a set of zero sub-patches $\hat{U} = \{\hat{u}_1, \hat{u}_2, \ldots, \hat{u}_k\}$ is firstly added. $\{U, \hat{U}\}$ is then mapped to corresponding position using $M$. Afterward, we project this combined set $\{U, \hat{U}\}$ into the embedding space $\{F, \hat{F}\}$, feed it into the encoder to obtain feature representations and reconstruct $\hat{P}$.

The mapping timing is the critical difference between the training and inference stages. The mapping is applied to the feature representations $F$ in the training stage. However, during the inference stage, the mapping is performed on the sub-patches before they are sent into the encoder. This difference is due to the use of positional embedding. In the training process, the positional embedding is applied before the erase operation because the sub-patches remain in their original positions at that stage. On the other hand, during the inference phase, the received un-erased sub-patches need to be mapped back to their original positions to incorporate the positional embedding effectively.

## D    LOSS

To minimize the difference between the original image $P$ and the reconstructed image $\hat{P}$, we adopt LPIPS Zhang et al. (2018), a well-known perceptual loss, and L1 loss as training loss. The final loss function is shown as follows:

$$L_1(x, y) = \frac{1}{N} \sum_{i=1}^{N} |x_i - y_i| \tag{1}$$

$$\text{LPIPS}(x, y) = \sum_{i=1}^{N} w_i \cdot d_i(x, y) \tag{2}$$

$$\text{Loss}(x, y) = \text{L1}(x, y) + \lambda * \text{LPIPS}(x, y) \tag{3}$$

Where $x$ and $y$ are two images being compared, $N$ is the number of layers in the feature extraction model, which is decided as VGG Simonyan and Zisserman (2015), $d_i(x, y)$ is the distance between the feature maps of the two images at layer $i$, and $w_i$ is a layer-dependent weighting factor. $\lambda$ is chosen as 0.3 in our experiments.

## E  ADDITIONAL QUANTITIVE RESULTS

### E.1  COMPARISON WITH OTHER SUPER-RESOLUTION METHOD

This section provides additional quantitive results for Easz compared with other super-resolution methods. From Fig. 8, it is evident that compared to the super-resolution method, Easz and the original image share more similar details, whereas super-resolution introduces more imaginative elements. The sys's-attributed to sys's direct pixel generation feature, making it better suited for use in conjunction with compression.

### E.2  QUANTITIVE RESULT OF EASZ ENHANCED COMPRESSOR

In this section, we provide additional quantitive results for the Easz enhanced compressor and the original decompressed image to show its effectiveness.

## F  DISCUSSION AND LIMITATIONS

### F.1  ADDITIONAL OPPORTUNITIES

**Online Finetuning.**In the edge-server scenario, conducting online training using real data presents practical challenges due to the server's unavailability of the original images. To enable online training on the server-side model with real data, it is crucial to meticulously choose the images for transmission and devise an efficient and well-designed method to maximize effectiveness.

Consequently, these features aid traditional compressors in improving the performance of subsequent image analysis tasks, particularly in scenarios with low-quality transmission. This observation opens an intriguing avenue for future research: How can we generate erased blocks more effectively to assist traditional compression methods better? This question could lead to exciting developments in the context of this paper.

**Semantic erase mask generation for specific scenarios.** A more promising direction for generating erase masks is to consider the semantic information of the image. This involves erasing only less significant regions. In this paper, such methods were not adopted to avoid additional computation on the edge. However, we still believe this is a viable direction for future research.

### F.2  LIMITATIONS

A fundamental assumption of Easz is that the server needs to be equipped with GPUs for efficient neural network processing. While common, these accelerators can increase costs and energy consumption. Moreover, if the reconstruction model is not well-trained, it may impact the overall performance of Easz. Additionally, on extremely low-power edge devices where other compression methods are not available, sys's compression capabilities, while still usable, might have limitations compared to more specialized alternatives.

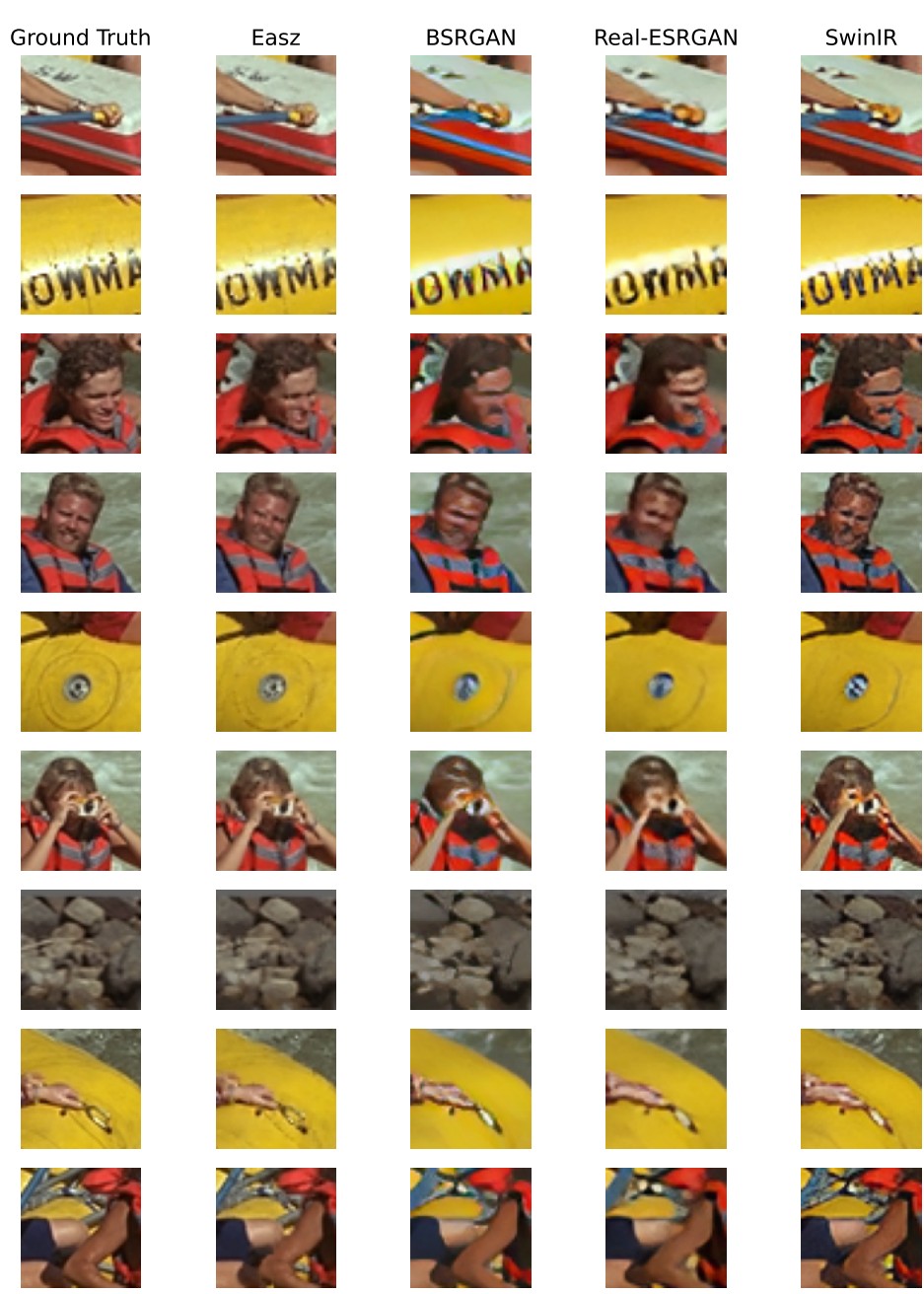

Figure 8: Quantitive result for Easz and other super-resolution methods.

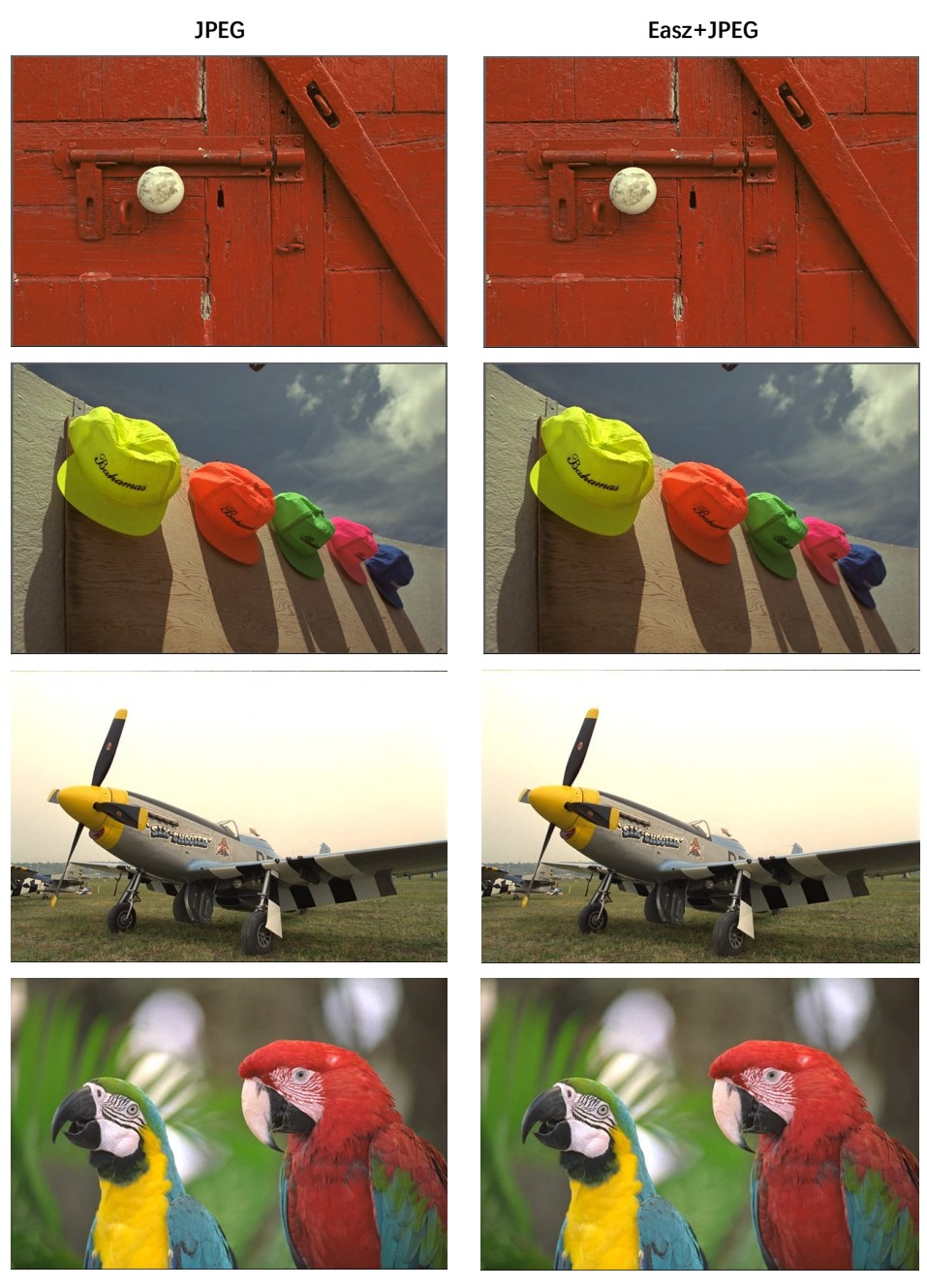

Figure 9: Quantitive result for Jpeg and Easz+Jpeg under compression ratio 20, erase ratio 20%.