# OpenReview forum: "Extending Flexibility of Image Coding Enhancement Framework for IoTs"
_ICLR.cc/2025/Conference — ICLR 2025 Conference Withdrawn Submission_

### Official Review · Reviewer_Rqp1 · 2024-11-01

**Soundness:** 2
**Presentation:** 2
**Contribution:** 2
**Rating:** 3
**Confidence:** 5

**Summary:**

This paper proposes an image compression enhancement framework for edge devices called Easz, which uses the ‘erase-and-squeeze’ technique to improve compression flexibility and efficiency.

**Strengths:**

It seems reasonable to downsample the original image to reduce the coding complexity at the transmitter side.

**Weaknesses:**

1. Under the experimental conditions where GT images are available, relying on no-reference metrics to assess the reconstruction quality will result in an incomplete assessment and make it difficult to clearly demonstrate the advantages of the method in preserving image details. Reference metrics can provide more direct quantitative comparisons in the presence of GT, so the inclusion of metrics such as PSNR and SSIM in the paper will make the experimental results more convincing and comparable.

2. The NN-based image compression models compared in this article all use autoregressive context modules, thus resulting in coding and decoding times that can be very long, and the authors need to compare them with parallel context modules (e.g. checkerboard [1])..

3. Higher complexity occurs on the receiver side using transformer reconstruction, especially for HD images. The complexity calculation in Supplementary Material Section B seems to be wrong. If the patch size is $N \\times N$, then the dim should be $d \\times N \\times N$, so the complexity of the model is not reduced.



[1] He, Dailan, et al. "Checkerboard context model for efficient learned image compression." Proceedings of the IEEE/CVF Conference on Computer Vision and Pattern Recognition. 2021.

**Questions:**

1. JPEG is inherently non-differentiable, how can the presence of JPEG compression be taken into account in training?

---

### Official Review · Reviewer_riKQ · 2024-11-02

**Soundness:** 2
**Presentation:** 2
**Contribution:** 2
**Rating:** 5
**Confidence:** 5

**Summary:**

This paper propose a paradigm to offer compression flexibility and efficiency improvement for edge-device scenarios. It propose a conditional uniform-based sampler for flexible image size reduction and reconstruction, as well as a lightweight transformer-based structure to redeuce reconstruction load on the receiver side.

**Strengths:**

- To avoid issues stems from adjacent sampled areas, this paper add constraints for row-based randoming sampling.

- This paper introduces an erase-and-squeeze method to enhance flexibility and efficiency.

**Weaknesses:**

- The proposed uniform-based sampler gives better results by adding constraints. However, it is still a rule-based sampling method that is unaware of the content. Why didn't you consider a learn-based sampler solution to improve the performance?

- The input of the reconstruction. Is the masking map also needs to be coded and transmitted? Is it also an input of the reconstruction network? I think these details should also be described in Figure 2.

- Settings about Table 1. Easz is a lossy compression method but other super-resolution methods such as SwinIR are free from bit cost constraints. Why does the proposed method Easz show such a performance improvement (larger than 3dB)? What are the testing settings(such as the input size) for the proposed method and the other methods?

- Some citations are neglected, e.g., the citation for the methods in Table 1 and Table 2. Please check about these problems.

**Questions:**

see the weakness

---

### Official Review · Reviewer_eHLc · 2024-11-02

**Soundness:** 3
**Presentation:** 3
**Contribution:** 2
**Rating:** 3
**Confidence:** 5

**Summary:**

The paper proposes Easz, a novel image compression framework designed for edge devices, addressing the limitations of existing NN-based compression methods. By introducing an erase-and-squeeze technique, Easz allows for flexible compression levels and efficient image reconstruction, making it suitable for various IoT applications.

**Strengths:**

1. The erase-and-squeeze method offers a flexible alternative to traditional uniform downsampling techniques, enhancing compression adaptability.
2. The framework is evaluated on a real-world testbed, providing credible evidence of its performance compared to existing compression methods.
3. Easz is compatible with existing compression algorithms, increasing its utility in practical applications.

**Weaknesses:**

1. The review of related work lacks depth, particularly regarding recent advancements in image compression techniques applicable to edge devices.
2. The reliance on GPU capabilities for reconstruction may limit the framework's applicability in low-power or resource-constrained environments.

**Questions:**

1. In the equation $G_u=\left\{g_u^0(i,j)=(i-1)/(h-1),g_u^1(i,j)=(j-1)/(w-1)\right\}$, is $g_u(i,j)$ a coordinate in the range $[0, 1]$? Can the authors clearly explain the difference between the symbols $X$, $\hat{X}$, $X_{squeezed}$, and $\hat{X}_{squeezed}$?
2. Does the erase-and-squeeze process disrupt the spatial structure between image pixels, negatively impacting conventional compression?
3. Do the authors compare their performance with related work that focuses on improving the quality of compressed images (e.g., RBQE) at the same bpp?
4. Why do the authors focus on optimizing L1 and LPIPS in the loss function while using unparameterized perceptual metrics instead of reporting performance metrics like PSNR, MS-SSIM, and LPIPS? How do the other models, SwinIR, realESRGAN, and BSRGAN, perform on these unparameterized metrics?
5. Is the erasure-extrusion process unique to each image? Additionally, does the quality of the reconstructed compressed image vary as a result? Does this pose a risk for downstream tasks?
6. How are the 2x downsampled images in Section 4.3 obtained? Are all three models—SwinIR, realESRGAN, and BSRGAN—67MB in size? The models I download are actually 64.16MB, 63.95MB, and 63.79MB, respectively.

---

### Official Review · Reviewer_AHiD · 2024-11-03

**Soundness:** 2
**Presentation:** 3
**Contribution:** 2
**Rating:** 3
**Confidence:** 5

**Summary:**

This paper proposes a new downsampling strategy for image compression on edge devices. The method achieves non-uniform sampling by erasing and squeezing the image blocks. In addition, the down-sampled image can be recovered using a transformer. Experiments show that the proposed method achieves superior rate-distortion performance (distortion measured by Brisque, Pi and Tres) with reduced computational complexity.

**Strengths:**

1. The paper is well written and easy to follow.
2. The idea of erasing and squeezing is interesting.

**Weaknesses:**

1. The proposed method is not suitable for image compression. The erasing process removes the image blocks randomly, which can lead to irretrievable losses of objects. An importance-based or smoothness-based method may be a better choice.
2. There is no comparison with uniform sampling methods, which is necessary to show the advantages and disadvantages of the proposed sampling method.
3. The non-reference distortion metrics used in this paper are not appropriate. As far as I know, the non-reference metrics can only measure whether the image is real, which has nothing to do with fidelity. So we can generate a completely different image at the decoder size with an image generation model without any bitstreams to get a high rate-distortion performance, which may be meaningless in applications.

**Questions:**

1. How does the proposed method perform when the distortion is measured using reference metrics such as PSNR and MS-SSIM?
2. How does the proposed method compare to uniform sampling methods?
3. Are there specific scenarios where we need to measure the distortion with non-reference metrics? Especially for image compression?

---

### Note · Authors · 2024-11-12

I have read and agree with the venue's withdrawal policy on behalf of myself and my co-authors.